# The microwave phase locking in Bloch transistor

Ilya Antonov[1], Rais S. Shaikhaidarov [1,2] ✉, Kyung Ho Kim [1,3], Dmitry Golubev [4,5], Sven Linzen[6], Evgeni V. Il'ichev[6], Vladimir N. Antonov[1] ✉ & Oleg V. Astafiev [1,7]

Recent experimental demonstration of the quantum coherent phase slip and current quantization in the superconductors, the fundamental phenomena dual to the coherent Cooper pair tunneling and voltage quantization (Shapiro steps), enables the development of a new quantum device, the Bloch transistor (BT). BT has a unique functionality: it can deliver quantized non-dissipative current to the quantum circuit. BT consists of two coupled Josephson Junctions (JJ) in the regime of coherent quantum phase slip. At the heart of the BT operation is a new mechanism for phase-locking the Bloch oscillations in JJs to microwaves via induced charge. The charge phase locking allows not only quantization of current but also gate voltage control of this quantization through the Aharonov-Casher effect. We study the operation of the BT and analyze its parameters. BT technology is scalable and compatible with other superconducting quantum devices, making it part of an emerging cryogenic quantum technology platform.

The discovery of coherent quantum phase slip (CQPS) in superconducting circuits has opened up a new direction of research and development[1]. The demonstration of the Charge Quantum Interference Device (CQUID) is an example, where a static charge of a small island between two weak links in the superconductor (the superconducting nanowires) altered the interference of magnetic fluxes tunneling coherently across the weak links[2]. The CQUID thus exploits the Aharonov-Casher effect. Another example is current quantization in superconducting nanowires or JJs under microwave (MW) radiation, the so-called Dual Shapiro steps[3–5].

The effect is a consequence of the coherent phase locking of the supercurrent with the MW. Further development is envisaged as the combination of two devices, the CQUID and Dual Shapiro steps. Such a device would have a new functionality, the delivery of gate-controlled non-dissipative quantized supercurrent to the quantum circuits. Conceptually, it would be called the Bloch Transistor[6]. Original works

on the CQUID and Dual Shapiro steps dealt with superconducting nanowires[2,3]. But from a practical point of view, devices with JJs, which act as CQPS elements, are more valuable because the fabrication of JJs is more reliable and controllable compared to superconducting nanowires[5].

To operate in the CQPS regime, JJ should have the Josephson coupling energy, $E_J$, and the charging energy, $E_C$ close to each other. $E_J$ and $E_C$ depend on the JJ critical current, $I_C$, and the capacitance, $C$

$$E_J = \frac{I_C \Phi_0}{2\pi}, \ E_C = \frac{e^2}{2C}, \tag{1}$$

where $\Phi_0$ is the superconducting flux quantum. Research with JJ is predominantly focused on two limiting cases of parameters: $E_J \gg E_C$ and $E_J \ll E_C$. They concern states where one of the conjugated quantum variables, either the superconducting phase $\varphi$ or the number

[1]Physics, Royal Holloway University of London, Egham, Surrey, UK. [2]National Physical Laboratory, Hampton Road, Teddington, UK. [3]Department of Physics and Astronomy, Sejong University, Seoul, South Korea. [4]HQS Quantum Simulations GmbH, Rintheimer Str. 23, Karlsruhe, Germany. [5]Department of Applied Physics, QTF Centre of Excellence, Aalto, Finland. [6]Leibniz Institute of Photonic Technology, Jena, Germany. [7]Skolkovo Institute of Science and Technology, Bolshoy Boulevard 30, Moscow, Russia. ✉e-mail: r.shaikhaidarov@rhul.ac.uk; vladimir.antonov@ntlworld.com

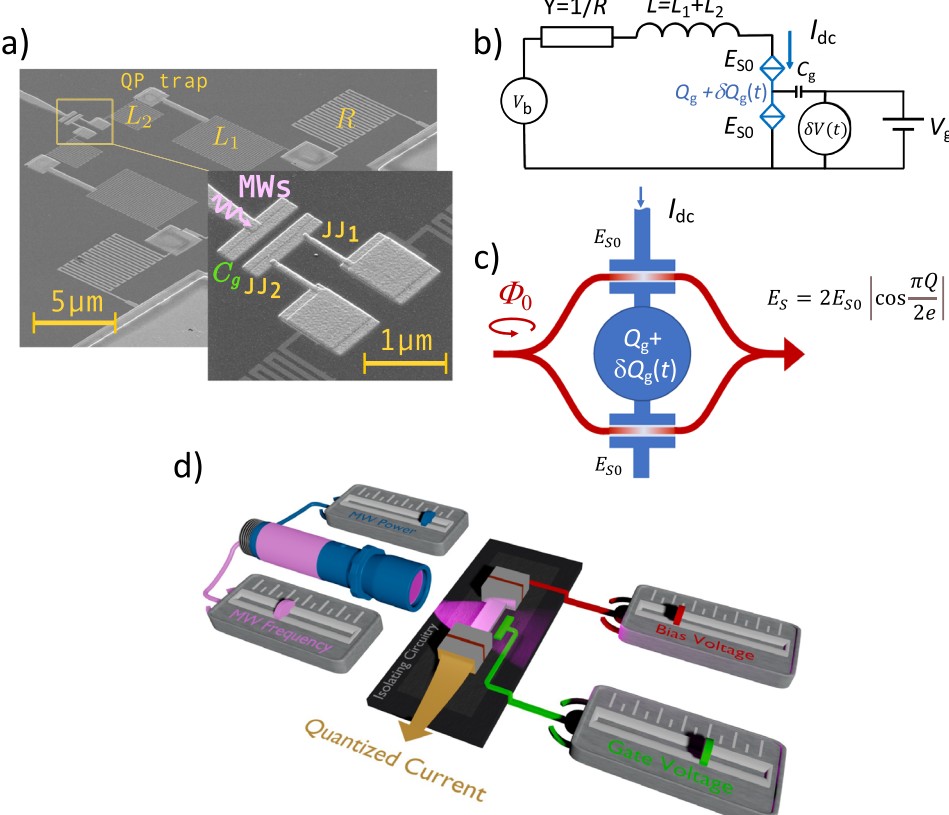

**Fig. 1 | Overview of the experimental sample. a** Focused Ion Beam image of the BT circuit. Two identical Al JJs separated by a small island embedded in the circuit with the super-inductors $L_1 + L_2 \sim 1.5\,\mu H$, resistors $R = 6.3\,k\Omega$ and quasiparticle traps QP. The MW is delivered to JJs by the gate electrode with capacitance $C_g$; **b** Equivalent electric circuit of the device; **c** Interference of the fluxons tunneling through the JJs; **d** Cartoon of the Bloch Transistor with four controls: the gate/bias voltage and the frequency/amplitude of the microwave.

of Cooper pairs $n$, is well defined. Examples of devices where $\varphi$ is a good quantum number are SQUID magnetometer[7] and Josephson voltage standard[8]. Their operation is based on the coherent evolution of $\varphi$ across the Josephson junction (JJ). The Charge Qubit operating with $n$ as a good quantum number, represents the opposite case[9]. The regime of JJs with $E_J \sim E_C$, is rather unexplored experimentally. The interest in this regime mainly concerned the parity issue and energy spectroscopy of the Cooper Pair Transistor[10–12]. Less attention was paid to the CQPS. The study of Bloch Transistor (BT) operation would shed light on the fundamentals of the CQPS phenomenon and its implementation to quantum devices with new functionalities.

In the current work, we combine two devices, the CQUID and Dual Shapiro Steps, and demonstrate current quantization that can be controlled with electrostatic gating. The phase-locking mechanism, which enables the current quantization, is unique in this device and different from that in devices with a single JJ. The phase-locking in our device is due to the oscillating charge on the island. This allows us to control the quantized current by gating through the Aharonov-Casher effect. In a system with single JJ, the phase-locking is the result of oscillating current, so gate control cannot be realized.

The experimental sample consists of two JJs separated by a small island with a gate electrode, see Fig. 1a, b. When the circuit is voltage biased and the MW is applied, the current quantization is developed. We control quantization with the voltage applied to the gate electrode through the Aharonov-Casher effect, Fig. 1c. Further experiments show that there are three other ways to control the quantized current: with the bias voltage, the amplitude, and frequency of the MW signal as shown in Fig. 1d. In the following text, for convenience, we refer to the device as BT.

## Results and discussion
### Experimental samples and model of the Bloch transistor
We carry out the experiments at extreme cryogenic temperatures ~ 15 mK. The BT is fully isolated from environmental electromagnetic (EM) noise. It is housed at the cold stage of the dilution refrigerator, and uses *dc* and *rf* circuitry, which are common for operation with the superconducting qubits, photon sources, resonators, etc. The *dc* lines connecting to the equipment at room temperature pass through the low-pass filters with a bandwidth of 1 GHz. The MW lines have attenuators of -60 dBm from the top to the bottom of the refrigerator, which strongly suppress arbitrary high frequency noise. Further filtering of the EM noise is done on the BT chip as shown in Fig. 1a. More details of the BT chip are explained later and in Supplementary Note 1. For *dc* measurement we use four-point methods with the symmetric differential amplifier. The measurement is a mixture of current and voltage bias schemes, where both voltage and current are probed independently of the bias $V_b$. The setup allows measuring $I - V$ curves with back-bending as in Fig. 2a. The principle scheme of the amplifier is given in the Supplementary Note 2[5].

The BT features two aluminum JJs separated by a small island, the isolation/screening circuit, and the combined MW feed line/ gate electrode. The on-chip isolation circuit has normal metal 15 nm-thick Pd resistors of $R = 6.3\,k\Omega$ and highly inductive 5 nm thick TiN meanders, $L_1 + L_2 \sim 1.5\,\mu H$. This high-impedance circuit strongly suppresses EM noise in the JJ leads, which is a requirement of CQPS operation[13]. The high inductance of the TiN film ensures a small footprint of the inductive element that minimizes parasitic stray capacitance and optimizes the total size of the device to < 100 μm². A TiN/Al/Pd sandwich, marked as quasiparticle trap (QP) in the figure,

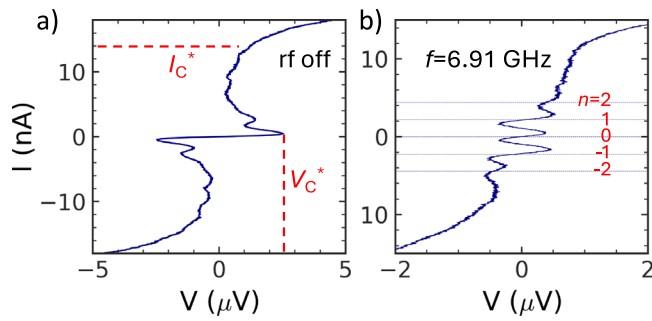

**Fig. 2 | Experimental $I-V$ curve without and with the MW. a** $I-V$ curve has current blockade below critical voltage $V_C^* = 2.5\,\mu V$. The apparent critical current is $I_C^* \sim 14\,nA$; **b** Current quantization under the MW of 6.91 GHz. The horizontal lines indicate the current corresponding to $I = 2efn$, $n = 0, \pm 1, \pm 2$. One can tune the BT to different $n$ by varying $V_b$.

relaxes the quasiparticles generated in the TiN meanders under MW radiation[5].

Two JJs of area $40 \times 90\,nm^2$ are of superconductor-insulator-superconductor type with normal resistance $R_n \sim 1.7\,k\Omega$ (measured in co-fabricated individual JJ junctions). From the value of $R_n$ we estimate $I_C = 186\,nA$, and $E_J/h \sim 92.5\,GHz$. The charging energy calculated from the geometrical capacitance of the JJ, 0.18 fF, is $E_{CJ}/h \sim 107\,GHz$. However, for a further estimate of the phase slip energy of the fluxon tunneling across the JJs, $E_{S0}$, one has to include in the calculation of the charging energy the parallel stray capacitance of the screening circuit, $\sim 1\,fF$[14]. We denote this energy by $E_C$. This reduces $E_C/h$ to $\sim 16.4\,GHz$. The combination of $E_C$ and $E_J$ enables us to calculate $E_{S0}$[5,15]

$$E_{S0} = \sqrt{\frac{8E_p}{\pi E_C}} E_p e^{-E_p/E_C}, \tag{2}$$

where $E_p = \sqrt{8E_J E_C}$ is the plasma energy[15]. By substituting the energies into the equation we get $E_{S0}/h \sim 0.55\,GHz$. When bias $V_b$ is applied to the BT, the current is blocked below the apparent critical voltage $V_C^*$ related to $E_{S0}$ by

$$V_C^* = \frac{(2\pi E_{S0})^2}{8e^2 R \delta I_T}, \tag{3}$$

where $R$ is the normal resistor of the screening circuit defined above, $\delta I_T \sim 1\,nA$, is the thermal noise of the screening circuit, see Supplementary Note 3. Above $V_C^*$ the BT has a branch with supercurrent. The latter is quantized when the MW is applied to the circuit. The effect is due to the phase locking of the MW with the Bloch oscillations in the JJs. The quantization is described by the equation

$$V(I_{dc}) = \sum_n J_n^2 \left(\frac{\delta Q_g}{2e}\right) V_0(I_{dc} - 2efn), \tag{4}$$

where $V_0(I)$ is the $I-V$ curve without the MW, see Eq. (S9) of Supplementary Note 3. In (4) $J_n(x)$ is the Bessel function of the $n$-th order and $\delta Q_g$ is the amplitude of the fluctuating charge on the island induced by the MW. The mechanism of phase locking is substantially different from that in the circuit with a single JJ[4,5]. In single JJ the MW locks the superconducting phase with the current $I_{ac} \cos(\omega t)$, while in BT it induces fluctuating charge at the island $\delta Q_g \cos(\omega t)$, see the full analysis in Supplementary Note 3. The form of Eq. (4) is similar to that of a single JJ when replacing the argument of the Bessel function from $\delta Q_g/2e$ to $I_{ac}/2ef$, where $I_{ac}$ is the amplitude of the MW.

The new mechanism of phase locking allows us to control quantization with the static charge $Q_g = C_g V_g$ induced on the BT island. The interference of the fluxons tunneling across the JJs and around the

static charge causes a modulation of the phase slip energy $E_S$ (the Aharonov-Casher effect)[2,16]

$$E_S = 2E_{S0} \left| \cos\left(\pi \frac{Q_g}{2e}\right) \right|, \tag{5}$$

In this equation, we assume that two JJs are identical. The modulation of $E_S$ propagates to $V_0(I)$ in (4), so that the plateaus of the quantized current should be periodically modulated.

**Operation of the Bloch transistor**

Three BTs have been studied. We present data for one of them. The $I-V$ curve resembles that of the usual JJ, but with the current blockade below the apparent critical voltage, $V_C^* \sim 2.5\,\mu V$, see Fig. 2a. It is smaller than $\sim 3.5\,\mu V$ calculated using (3). The difference can be attributed to the uncertainties of the stray capacitance and thermal noise current used in the calculation. We should note that BT operates in the limit of strong noise $eV_C^* < k_B T$. At small bias the differential resistance reaches ~20 k$\Omega$.

Above the apparent critical voltage, the $I-V$ curve has the supercurrent branch with the apparent critical current $I_C^* \approx 14\,nA$. Then the experimental points follow the Ohmic law. The $I_C^*$ is smaller than the critical current $I_C \sim 100\,nA$ of co-fabricated JJs which are not embedded in the screening circuit, see Supplementary Note 4. One can relate $I_C^*$ to Zener current $I_Z = (\pi E_J/16 E_{CJ}) I_C$[4,5]. By substituting $E_J$ and $E_{CJ}$ we get $I_Z \sim 17\,nA$, which is fairly close to experimental $I_C^*$. The resonances are visible on the $I-V$ curve. The most evident resonance is between 2 nA and 4 nA. We relate these resonances to the rectification of white noise by the isolation $LC$ circuit of the BT that has its own resonance frequencies. We confirmed that in the BT with reduced inductance, the resonances shift to higher frequencies.

The current plateaus are developed at the $I-V$ curve when the MW is fed to the BT through the gate electrode, see Fig. 2b. Five plateaus, $I_n = 2efn$ with $n = 0, \pm 1, \pm 2$, are visible when the MW frequency is $f = 6.91\,GHz$. The differential resistance of the quantized plateaus is smaller than the "blockade" resistance without the MW. It reaches only 1 k$\Omega$ in the centers of the plateaus. In the BT under study, quantization is present between 6.7 GHz and 10.4 GHz, see Supplementary Note 5. The maximum quantized current is ~6.6 nA. The quantized current can be accurately controlled by means of four handles: gate and bias voltages, the amplitude and frequency of the MW. The prime control is done with the gate voltage. The application of $V_g$ periodically changes the slopes of the current plateaus, so that the current periodically deviates from the quantized value. The effect is clearly seen in the differential resistance $dV/dI$, see 3D plot in Fig. 3a. The peaks of the differential resistance are positioned in the centers of the current plateaus $I_n = 2efn$. They are only ~1 k$\Omega$ (the maximum of $dV/dI$ at low bias without the MW is ~20 k$\Omega$). Due to the low floor of $dV/dI$ under the MW, the gate voltage modulation is as high as 40% (it is only 14% without the MW). As we discussed above, the modulation is due to the control of the interference of the fluxon tunneling across the JJs (the Aharonov-Casher effect). The modulation period $\Delta V_g$ in Fig. 3 is $e/C_g$. We confirm the period using experiments with the Single Electron Transistors (SET) of the same geometry as the BT. The Coulomb Blockade Oscillations of the SET are predominantly $e/C_g$-periodic, except in some special cases[17]. The experimental curves of the SET are shown in Supplementary Note 6. The modulation period of BT is half that expected from (5). We relate the effect to the poisoning circuit with the quasiparticles generated in the inductors $L_{1,2}$ by MW. The appearance of quasiparticle poisoning is different in the BT and in the CQUID[2]. In the BT the time constant of the probing $dV/dI$ largely exceeds tunneling and relaxation times of the quasiparticles. Because of this, the effect is seen as the modulation of the differential resistance with the period corresponding to the electron charge $e$ after averaging two states with different parities. This averaging reduces the depth of

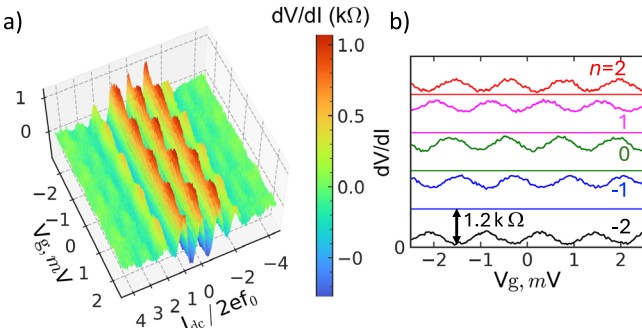

**Fig. 3 | Gate control of the BT. a** The intensity graph of the differential resistance $dV/dI$ vs normalized current $I_{dc}/2ef$ and gate voltage. The peaks of $dV/dI$ are at the centers of the quantized plateaus $I_{dc} = 2efn$. They are periodically modulated with the charge $e = V_gC_g$ induced at the island between the JJs; **b** Stack of cross sections of the intensity graph taken at fixed $I_{dc}/2ef$ corresponding to different $n$. Each curve has the absolute value of $dV/dI$, between zero and 1.2 k$\Omega$. Zeros of $dV/dI$ of each curve are shown as the solid line of the corresponding color. There is a phase shift of the gate modulation between different $n$.

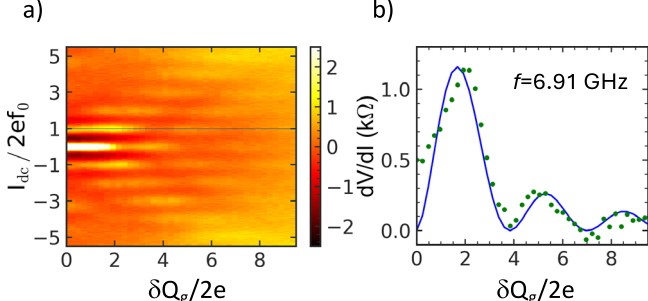

**Fig. 4 | MW control of the BT. a** Intensity plot of the differential resistance $dV/dI$ at different bias and $\delta Q_g/2e$. The bright peaks are located at the quantized current plateaus, $I_{dc} = 2efn$; **b** Cross section of $dV/dI$ at $I_{dc} = 2ef$ (green dots). The solid blue line is a fit of the experimental data with the square of Bessel function $J_n^2(\delta Q_g/2e)$ with $n = 1$.

the modulation also by half in BT. In the CQUID the probing of the qubit state is faster than the evolution of the quasiparticle. As a result, one can see simultaneously two states of the qubit with different parities, each with $2e$ periodicity. The gating effect is weakly pronounced in the direct $I - V$ curve, see Fig. S6 of Supplementary Note 5.

One can also note that the position of the peaks in $V_g$ shifts linearly with $I_{dc}$. It is clearly seen in Fig. 3b, where the modulation curves at different current plateaus ($n = 0, \pm 1, \pm 2$) are compiled together. The phase of the oscillations is consistently shifted with the bias. It can be related to the presence of a small capacitance of the JJs themselves: the voltage $V$ across the BT may induce the extra charge at the island, which is added to that induced by the $V_g$, and change the oscillation phase.

Finally, to enable the Aharonov-Casher effect, the kinetic capacitance in the circuit $C_{kin} = e/\pi V_C$ ($V_C = \pi E_S 0/e$) should be greater than $C_g$[18]. This condition is satisfied in our devices: $C_{kin} = 7.1$ fF and $C_g = 0.134$ fF.

One can also vary the quantized current with the bias voltage, $V_b$. The effect is easy to perceive from the $I - V$ curve of the BT, see Fig. 2b. One can move from one current plateau to another by varying $V_b$. Such a control is possible because in our measurement scheme the relation between the voltage applied to the circuit, $V_b$, and voltage measured across the BT itself, $V$, is not linear. However, the current-voltage relationship does not have one-to-one correspondence and depends on the history. The full range of control is limited by the apparent critical voltage $V^*_C$, which is $\pm 2.5\,\mu$V in the current BT.

The MW control of the BT with the amplitude and frequency of the MW, $\delta Q_g$ and $f$, comes from (4): the width of the current plateaus is modulated with the MW amplitude while the current value itself is proportional to $f$ as $I = 2efn$. The effect is well pronounced in differential resistance $dV/dI$, see Fig. 4a. In the figure $I_{dc}$ and $\delta Q_g$ are normalized to the quantized current step $2ef$ and charge $2e$. The light colors at $I_{dc}/2ef = 1, 2, 3, 4$ correspond to the maxima of $dV/dI$ at the centers of the quantized current plateaus. When $I_{dc}$ is fixed to the quantized value $2efn$, the peaks of $dV/dI$ follow $J_n^2(\delta Q_g/2e)$ dependence, see Fig. 4b. The curve is approximated with Eq. (S10) of Supplementary Note 3. The variation of $\delta Q_g$ deviates the current from the quantized plateaus $2efn$.

The BT can potentially deliver non-dissipative quantized current to the quantum circuit with high accuracy. Such functionality is useful for quantum circuits where the dissipation is an issue and the operation is carried out with precise tiny signals.

## Further optimization and application

The accuracy of BT's current quantization is currently limited. For example, a precision better than 1 ppm is needed for the current standard. A fair analysis of the quantization accuracy and ways to improve it are given in ref. 19. One focus of the development is the thermal noise of the resistors in the screening circuit. MW applied to the BT heats the normal resistors, so that the fluctuating current $\delta I_T$ is activated. We estimate this current as ~1 nA, which is comparable to the amplitude of the quantized current itself. One can either optimize the coupling of the MW to the JJ so that a weaker external MW can be used[18]. Alternatively, the on-chip JJ generator of the MW with good impedance matching can be used[20]. One can also increase $E_S$ by designing JJ with higher $E_J$ and $E_C$, while keeping $E_J/E_C \sim 1$. However, the nano-fabrication imposes the limit. To increase $E_C$ one has to decrease the capacitance of JJ. The latter requires reducing the lateral size of the JJ. It is challenging to fabricate JJ smaller than $40 \times 40$ nm$^2$ in a controllable way. One can also explore more effective cooling of the BT chip. Recently, immersion cooling of superconducting resonators in the $^3$He bath below the common plateau of 50 mK was reported[21]. The immersion cooling technology is compatible with the BT's operation. However, it is a further complication towards the practical application of the device.

Optimization can also be done in the BT control circuit. In our measurements, we use a mixture of the voltage- and current- bias scheme[5]. Ideally, the pure voltage bias scheme should be used to properly bias the BT.

There are two straightforward applications of the BT in metrology and quantum coherent circuits. The BT can be used as the absolute quantum current standard in metrology. We recently reported on developing the metrological chip that contains voltage and current standards[14]. The BT can be naturally accommodated on this chip. The second application is envisaged in quantum circuits. BT can deliver the current to the control flux line in the qubit with the SQUID loop. For this application, BT has unique combinations of local on-chip design, compatibility of the fabrication technology with superconducting qubits, and a reduced back action on the quantum circuit because of the non-dissipative nature of the quantized current. The latter is important for ensuring the long decoherence time of the quantum circuit operation.

In summary, we demonstrate a new mechanism of phase locking with the MW in operation of the Bloch Transistor. Two phenomena, the Dual Shapiro Steps and Aharonov-Casher effect, are present in the BT.

The operation of the BT is unique, as the non-dissipative current quantization is engaged with the fluctuating charge in the double JJs circuit. The maximum amplitude of the quantized current is ~ 6.6 nA. The BT features four controls to set the current level, deactivate the current locking, and modulate the current amplitude. It can be part of a coherent quantum circuit that delivers a non-dissipative current. The technology of BT fabrication is scalable and compatible with that of other superconducting quantum devices. We believe that BT can be an essential part of the new cryogenic quantum technology platform. However, further improvements should be implemented to make the device more accurate and resistant to noise.

## Methods

The fabrication of experimental samples includes four processes: Ti/Au (10 nm/80 nm) contacts for bonding, aluminum JJs, TiN (5nm) super-inductors and Pd resistors (15 nm). Super-inductors are prepared with ion etching of the ALD-grown TiN film in $CF_4$ plasma. The inductance per square is ~ 2 nH. Two TiN meanders have inductances 1.15 μH and 0.34 μH. Pd and Al are deposited by thermal evaporation. The Pd resistors have $R_{square}$ ~ 10$\Omega$. Aluminum JJs, Al/AlO$_x$/Al, are fabricated with the shadow evaporation technique. The normal resistance of the JJ is $R_N$ ~ 600$\Omega$ recalculated for junction size 100 × 100 nm.

Low-temperature experiments are conducted in a dry dilution refrigerator with a base temperature of 15 mK. The thermo-coaxial cables are used for *dc* lines from room temperature to 15 mK. They are thermalised at 50 K, 4 K, 800 mK and 15 mK. Twisted pairs are used for electric circuitry at the base temperature. The *dc* signals pass through cascade of LTCC low pass filters with a stop band from 80 MHz to 20 GHz before reaching the BT. The MW lines have attenuators at different temperature stages with a total value of -60 dB. The $I - V$ and d$V$/d$I$ curves are taken using the differential amplifier at room temperature.

There are events of arbitrary phase jumps in $dV/dI$ modulation with the gate voltage. The occurrence of them increases when the MW is applied. Their typical time constant is few seconds (the transconductance of the BT with frequent switches is shown in Fig. S9 of the Supplementary Note 7). It is highly probable that we encounter a fluctuating charge in the materials. The stability of BT against switches can be improved by annealing BT to 4 K, and cooling it back to the operation temperature of 15 mK. The graph in Fig. 3 is compiled after the annealing procedure.

## Data availability

The data generated in this study have been deposited in the Open Science Framework repository. They can be obtained without any restriction at (https://osf.io/a8nhp). Additional information, experimental curves, and schemes are also provided in the Supplementary Information.

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

## Acknowledgements

This work was supported by Engineering and Physical Sciences Research Council (EPSRC) Grant No. EP/Y022637/1, European Union's Horizon 2020 Research and Innovation Program under Grant Agreement 20FUN07 SuperQuant. K.H.K. acknowledges support of MSIT grant IITP-2025-RS-2024- 00437191.

## Author contributions

O.V.A., R.S.S., E.V.I. and V.N.A. conceived and supervised the experiments. R.S.S., K.H.K., S.L. and I.A. fabricated BT and performed measurements at low temperatures. All authors contributed to the analysis and simulations of the data. E.V.I., D.G., I.A. and V.N.A. wrote the manuscript, and all authors contributed to editing the manuscript.

## Competing interests

The authors declare no competing interests.
