## [Transparent Peer Review file · Nature Communications]

The microwave phase locking in Bloch transistor

Corresponding Author: Professor Vladimir Antonov

Version 0:

Reviewer comments:

Reviewer #1

(Remarks to the Author)

Comments on the Manuscript "Bloch Transistor for Cryogenic Quantum Electronics"

This manuscript presents a superconducting device comprising two Josephson junctions in series, which form a small superconducting island and are embedded in a high-impedance circuit that ensures charge quantization and isolation from external noise. This device is referred to as a Bloch transistor. While the topic has potential interest for quantum electronics and metrology, I believe that the current version of the manuscript does not meet the standards of Nature Communications and cannot be considered for publication for the following reasons:

1. Unclear Main Achievement

The main contribution of the paper is not clearly defined. Is the primary claim the Bloch transistor itself? The electrical control of dual Shapiro steps? The potential application for controlling quantum devices? The introduction and title emphasize the potential application of the device for controlling quantum systems; however, there is no data presented that demonstrates how the Bloch transistor might be superior to existing control methods. Can the authors provide evidence that it is more stable, more precise, or less noisy?

In my opinion, the authors need to choose whether they want to publish a physics paper or focus on presenting a device with potential applications. Currently, the title and the introduction suggest an emphasis on applications, yet the performance of the device is not addressed or compared to existing alternatives. If the authors intend to explore the physics of the Bloch transistor, they should significantly rewrite the paper and clearly explain what makes the physics of this device novel and distinct.

2. Significant Overlap with Previous Work

The manuscript exhibits substantial overlap with previously published work by the authors, particularly:

- Nature 2022, s41586-022-04947-z

- Nature Communications 2024, s41467-024-53600-y

The only significant modification seems to be the transition from a single junction to a series of two (i.e., the transistor configuration), which does not, in itself, constitute a sufficiently novel contribution. The only data that could be interpreted as demonstrating a new feature is presented in Figure 3, which relates to the gating of the superconducting island. However, this figure is confusing: it presents only differential conductance, rather than the actual current, which is more relevant for transistor operation. Additionally, the authors mention that "the current plateaus dissolve and reappear with the MW amplitude," but the influence of the gate voltage on the main features (as shown in Figure 2) is not adequately presented. The discussion primarily focuses on current metrology, similar to the authors' previous works.

3. Ambiguity Surrounding the Biasing Scheme

The measurement setup is not sufficiently described in the main text or the supplementary material. This omission is particularly problematic regarding the IV curves shown in Figure 2. It is likely that these curves include a resistance correction to display "back-bending" features, which gives the misleading impression that multiple current values correspond to the same voltage—a concerning trait for a transistor or current source.

In reality, the uncorrected data would likely show a single current for each applied bias voltage, with quantized current values as the bias voltage varies. In this context, the role of the gate voltage is unclear. The authors should clarify the operational advantages of using the gate voltage in comparison to the bias voltage and explain how gating enhances or differentiates this device from their previous designs.

****Two related notes:****

- Presenting the uncorrected IV data would be beneficial in evaluating the actual extent of the current plateaus, specifically how many data points contribute to each step. This is crucial for assessing the precision required for voltage control in practical applications.
- What do the authors mean by "One can tune the BT to different n by varying V_b " (Figure 2 caption)? The current form of this figure indicates that multiple currents can be obtained for the same V_b . An uncorrected IV curve would likely demonstrate more clearly that different n values can be accessed by varying V_b .

4. Incomplete Literature Review and Missing Comparisons with Prior Superconducting Transistors

The manuscript does not adequately acknowledge a significant body of literature related to superconducting transistors and other gate-modulated devices. Among the relevant references that have been omitted are:

- PRL 92, 066802 (2004)
- PRL 95, 206807 (2005)
- Nature Physics 12, 825–829 (2016)
- PRL 98, 216802 (2007)
- PRB 99, 220504 (2019)
- PRRResearch 4, 013038 (2022) and the references therein.

A meaningful comparison with at least the most technically similar devices is essential. Furthermore, to justify the significance of this work, the manuscript should clearly articulate how the presented Bloch transistor offers tangible improvements over these earlier proposals and demonstrations, which is currently lacking.

5. General Style and Presentation Issues

The manuscript lacks clarity and the main claim is not properly defined. The structure could be significantly improved to better guide the reader through the novelty of the device and present the experimental results in a more intuitive and coherent manner.

Conclusion

In its current form, this work appears to be a largely incremental advance over the authors' previous studies. The lack of a clear conceptual motivation, combined with insufficient data presentation and the absence of comparison with existing literature, makes it difficult to justify publication. Significant revisions, both in terms of scientific content and clarity, would be required to bring the manuscript to Nature Communications standard.

General Remarks on the "Bloch Transistor for Cryogenic Quantum Electronics"

1. Quantum Dissipative Nature of the Device: The authors describe the device as purely quantum and non-dissipative in the abstract. However, they discuss operating the junction in the "Bloch regime," which actually involves a dissipative environment.
2. Junction Parameters: The junction parameters appear to be quite high, and they seem to shift unexpectedly at various points throughout the paper. For example, using the provided values for E_J and E_C to calculate the plasma frequency yields a frequency around 200 GHz, which is significantly higher than the aluminum gap. The paper would benefit from a clearer explanation or justification of these parameter choices.
3. Calculation Accuracy: Several calculations in the paper are not performed with sufficient care, and noticeable errors are present. For instance, based on the given E_J , the critical current I_C of a single Josephson junction should be approximately 180 nA. However, the paper mentions a value of 100 nA, despite initially defining I_C as the critical current of one junction. Additionally, when calculating the apparent current I_C^* , the authors use the formula $I_C^* = (\pi E_J) / (16 E_C) * I_C$. Plugging in the values results in approximately 64 nA, while the paper states 14 nA, which is unexpected.
4. Missing Error Bars and Unexplained Behavior in Fig. 4: The plot in Fig. 4 (the one on the right) lacks error bars, making it difficult to assess the quality of the fit. Furthermore, the differential conductance (dV/dI) does not approach zero at the origin of the applied AC current, yet the paper does not describe or explain this inconsistency.
5. Insufficient proofreading of the manuscript: for example, the current title of the supplementary material is "Quantized Current Steps Due to the Synchronization of Microwaves with the Bloch Oscillations in Small Josephson Junctions."

(Remarks to the Author)

Review on the manuskript: Antonov et al., "Bloch transistor for cryogenic quantum electronics"

Bloch transistor (BT, also: "Bloch electrometer", "Cooper pair transistor", etc.) - two small, closely spaced Josephson junctions, in which the charge on the island between the junctions can be varied by an external gate voltage – has been extensively studied since late 80th; theory and experiments published in a few tens of papers. The charge dynamics in this circuit reflects the interplay of Josephson and Coulomb effects, which, in the case of the dominating Josephson energy and low-bias operation, can be reduced to that of a single small Josephson junction having a gate-controlled Josephson coupling $E_J = E_J(Q)$, where Q is a gate-induced charge of the island. A current I through a small Josephson junction is related to the so-called Bloch oscillations which are oscillations of voltage across the junction with a period proportional to $2e/I$. It has been a long-standing but challenging idea to lock the Bloch oscillations to a microwave drive, and so to generate steps of constant current, dual to Shapiro steps in the Josephson voltage standard. For very dissimilar Bloch and the drive frequencies, the resonant mixing of Bloch oscillations with the probe signal was implemented in a reflectometry setup published in Nguyen et al., PRL 99, 187005 (2007). A remarkable advantage of the experiment under review is that such an effect is reported now for matching frequencies and registered in form of DC current steps (so-called dual Shapiro steps) in the IV-curve of a Bloch transistor embedded in a high-impedance biasing circuitry.

The reported achievements mark a recent progress in the field after publishing Refs. [8-10] as well as other important papers during the past few years. Correspondingly, this new study can be understood at best in the tight context of the preceding works, but it does not represent a crucial "game changer" in the field. The data are definitely of substantial interest for scientists involved in the studies of small Josephson junctions. However, at the present form the manuskript should not be published and it needs, in my opinion, to be conceptually reworked.

Below I will list a few points of major and minor criticism, in the hope it could help to produce a high-quality paper of profound interest.

1. The title of the manuskript ending by "... for cryogenic electronics" implies a high degree of universality of the message, but up to now there has been only a few applications, envisaged solely in metrology, related to the fundamental current-to-frequency relation, $I = 2e \times f$, which are in focus of the present study.

2. In the Introduction, there is a very scarce review of previous works related to Bloch transistor, despite the decades-long and extensive scientific context behind. The cited few works are mostly related to superconducting microconstrictions or single tunnel junctions and the way to interpret them in terms of coherent phase slips, but not to BT.

3. In the Introduction, a need for low-noise current sources for quantum devices is used as motivation for the work. On the other hand, it appears obvious (and is repeated several times in the manuskript) that the transistor itself requires four external signals for operation, which would certainly be the same way not free of noise as any other. The effect of noise is known to increase the linewidth of Bloch oscillations leading to fluctuations of the nominally quantised DC current component. Moreover, the output signal across the dual Shapiro steps will necessarily involve, along with the DC term, also a long set of harmonics as well as the trace of the driving signal itself. Another point: proper operation of BT was shown in the paper to happen in a very narrow, sub- μV bias range. This leads to extremely high sensitivity of BT to the back-action processes in the possible load. As a quantum circuit, the load will be typically controlled and manipulated by GHz signals thus producing inevitably strong disturbances to BT. Neither the accuracy nor the current level of the dual Shapiro steps presented in the paper are superior to those previously reported for microconstrictions/single junctions in Refs. [8-10]. The argument is then open why using BT instead of much simpler single junction circuits. To summarize, I would not expect any impact of Bloch transistors as a current source for quantum circuits supporting the statements declared.

On the other hand, I would like to stress once again the scientific interest and high relevance of the results obtained. In particular it is so, because of an intriguing, in fact, parametric - due to the dependence $ES(C_g \times V_g)$ - mechanism of coupling the microwave drive and the Bloch oscillations in the setup under study.

4. Starting at the introduction and following in the next section, the phenomena are treated by focussing on the coherent flux tunneling and Aharonov-Casher effect, which is a usual treatment of superconducting CQPS microconstrictions. This viewpoint is dual to that describing the coherent Cooper pairs transport in single charge transistors and by far not intuitive, nor insightful for a wide-scope scientific journal. Just to note it briefly, for small Josephson junctions there has been essentially similar, but more direct formalism based on charge-phase Hamiltonian adopted by Averin, Zorin, and Likharev in Sov. Phys. JETP, 61 (2), 407 (1985) and further developed towards Bloch transistor in Zorin, PRL, 76, 4408 (1996). Whatever approach is used, it should be self-explaining in respect to the phase locking mechanism. In BT the mechanism is expected to be different from that of a single junction (or QPS junction), because it does not involve an RF driving signal injected in parallel to the DC bias, but rather the parametric gate-modulation of the Josephson coupling strength (cf. discussion around Eq. 4 in the manuskript). Under this condition, the equation (3) developed for a single junction may not be applied directly to BT and needs to be replaced by a more dedicated expression.

In this respect, it is also not clear (neither defined explicitly in the manuskript) which quantity is denoted by I_{ac} ? Due to the strong Josephson coupling, the individual junctions in BT can not be considered as discrete components experiencing individual gate injection, but even if they could, the sign and value of two injected currents would be two deviating parameters in stead of one I_{ac} .

To my understanding, developing a model of dual Shapiro steps in BT based on the parametric driving effect would be of significant importance for understanding the experimental findings reported.

Finally, the logical structure of the manuskript would benefit of a more concise model description without scattering the details across sections I, II, III as now.

A few minor points are listed below:

1. In Fig. 1 the bias voltage (and the biasing layout generally) is drawn one-sided and open-ended. This deviates much from the usual (and de-facto implemented on chip) circuit design of BT and, strictly speaking, is unphysical.
 2. In Fig. 1 the quantities L_1 and L_2 are mentioned but not labeled.
 3. Page 2, left: «Inverse Shapiro Steps» is not widely called so in literature. The most used term is “dual Shapiro steps”.
 4. Bottom paragraph of Page 2, left: Passive circuit components usually constitute the EM environment, but they do not belong directly to BT.
 5. It is not clarified in the text in which way the tunneling energy of the fluxon E_S is related to the junction current. Surely, one can learn more from Ref. [8], but that would render the main text being not self-explaining. See also above the comment to the model.
 6. Equation (2) is neither derived here, nor referred to a literature source. From Ref. [13], the expression for the plasma frequency is indeed available, but not the expression for E_S . From the text flow, it is not quite clear, either the quantity E_S is related to a single junction or to BT (“In the BT under study $E_S/\hbar \sim 15$ GHz”)
 7. Page2, right: “Finally, this parameter determines the accuracy of the current quantization and the operating temperature.” This statement is not clear and should be explained. It is also not obvious how the temperature depends on E_S .
 8. “The BT delivers quantized current when it is voltage-biased,..” This statement, if assumed to be valid for a gate-voltage driven BT, should be justified or referred to literature (theory?). Furthermore, it apparently contradicts to the requirement of high-impedance biasing circuit for observation of Bloch oscillations. In case of pure voltage bias, no voltage oscillations can just appear across the double junction.
 9. The parameter V_C is used before it is defined in the text.
 10. Equation (4) is is neither derived, nor referred to a literature source.
 11. Fig.2, caption (also in the text): “One can tune the BT to different n by varying V_b .” From Fig. 2 it is not so obvious that one can reach, say, the step at $n = 0$ when starting from other current steps and varying the bias voltage only. As shown in Fig. 1 and in supplementary information, the IV-curve was measured in a four-point layout utilising a current bias and a voltage measurement procedure.
 12. The numeric values of some quantities seem not to be quite consistent. For example, cf. “capable of delivering a quantized current up to 10 nA.” at the end of the Introduction, 14 nA in conclusions and “quantization is present between 3 GHz and 20 GHz” in Section III. On the contrary, the resistive (Landau-Zener) branch in Fig.2 sets on at about ~ 5 nA which is even lower. The resistance scale of ~ 1 kOhm in Fig. 3 is much lower than stated: “The differential resistance of the quantized plateaus is below the “blockade” resistance of 20 k Ω ”. Similar deviations can be found in respect to the bias voltage estimates up to 10 μ V, while the plotted values are all much lower. The same for V_g .
 13. Page 3, right: The explanation of the gate voltage dependence related to Fig.3 in terms of fluxes and Aharonov-Casher interference is not directly insightful for interpreting the differential resistance of RF-driven BT, especially under the large gate signals sweeping over several periods, $Q > n \times 2e$.
 14. For Fig. 4 neither bias nor dc gate voltage values are indicated (even if maybe not so substantial for the pattern).
 15. Since the microwave power values are provided in the normalized form as I_{ac} , it is difficult to conclude how strong the signal was in terms of $Q/2e$ and either there was an effect of mutual synchronisation of two Bloch oscillations: the one along the bias path and the other induced by the gate to the island as a box (similar to Nguyen et al. cited above).
- Finally, I would like to encourage the writing authors to further optimize the text flow towards easier understanding of physics behind the experiment.

Reviewer #3

(Remarks to the Author)

When a Josephson junction is embedded into a high-impedance environment, its physics becomes dominated by effects of electric charge discreteness. One such effect is Coulomb blockade. The blockade impedes the flow of supercurrent, and makes the junction display insulating (as opposed to superconducting) signatures. Another effect is Bloch oscillations. These are the oscillations of voltage across the junction biased by direct current I . The Bloch oscillations occur with a frequency $f = I / (2e)$ that only depends on the value of the current and unit of charge $2e$. It is the latter effect that is the cornerstone of the authors' work.

Specifically, the authors introduce a device called a Bloch transistor. In this device, two Josephson junctions are connected in series, and an island between them is capacitively coupled to a gate electrode. The high-impedance environment is provided by on-chip resistors and inductors. The two-junction structure allows one to control not only the frequency, but also the amplitude of Bloch oscillations, through the Aharonov-Casher effect (i.e., by varying the gate voltage). The authors propose to use the Bloch transistor as a source of quantized current. This application can be achieved by synchronizing Bloch oscillations with external microwave radiation. The microwave frequency f sets the quantized values of the current, $I = n * 2ef$.

To verify the feasibility of this application, the authors measure the current-voltage characteristics $I(V)$ of the device in the presence of microwaves. They observe that $I(V)$ develops a set of quantized steps, at the expected values of I (the so-called dual Shapiro steps). They also show that the sharpness of the steps – and thus the precision of the quantization – can be controlled by varying the gate voltage; this observation is in qualitative agreement with the Aharonov-Casher effect.

Overall, the authors' results make sense to me. The data convincingly demonstrates the dual Shapiro steps in the Bloch transistor; it also illustrates the possibility of controlling the parameters of the steps using gate voltage, and frequency and amplitude of microwaves. Unfortunately though, I do not get an impression that the results represent a significant advancement over the previous works. I also have questions and concerns regarding the authors' analysis. Therefore, I

cannot recommend publishing the article in Nature Communications at this stage. I elaborate on the reasons for my assessment below.

To begin with, let me note that both the dual Shapiro steps and the Aharonov-Casher interference were previously demonstrated. In the former case, this includes a series of works by the same authors on superconducting nanowires [Nature 608, 45 (2022)] and Josephson junctions [Nature Communications 15, 9326 (2024), Appl. Phys. Lett. 125, 122602 (2024)], as well as works by other authors [Nature Physics 19, 851(2023); Nature Communications 15, 8726 (2024)]. Aharonov-Casher interference – i.e., the interference between the coherent phase slips – was also observed in a number of contexts, including that of the Josephson junction arrays [Phys. Rev. B 85, 094503 (2012), Phys. Rev. B 85, 024521 (2012)] and disordered superconductors [Nature Physics 14, 590 (2018)]. The existing literature makes me think that the only novel aspect of the present manuscript is a direct verification of the connection between the two effects. The connection is by no means unexpected: in theory, the Aharonov-Casher interference sets the total amplitude of the phase slip in the device; it is the latter amplitude that determines the width and the slope of dual Shapiro steps. The authors present data that is qualitatively consistent with this expectation. The data will, certainly, be of interest to specialists that closely follow the development of “dual” superconducting electronics. In my opinion though, the advance is too incremental to merit a publication in Nature Communications, and would better belong in a more specialized journal.

In addition to the high level remark above, I have several specific questions and comments.

Questions and concerns:

– A key parameter in the theory of charge discreteness effects in a Josephson junction is the environment impedance Z (and, specifically, how Z compares to the resistance quantum $R_Q = h / 4e^2 = 6.5k\Omega$). Can the authors estimate the value of Z at relevant frequencies for their circuit? Is it simply determined by the two Pd resistors in series? Do the TiN inductances produce a significant contribution to Z ?

– I don't understand the status of Eq. (4). Is this equation correct? If so, what limit does it apply in? In its present form, Eq. (4) contradicts my naive expectations. Indeed, for the case in which the phase slip amplitudes at the two junctions were the same, I would expect the relation to read $E_S = E_{\{S0\}} |\cos(\pi Q/2e)|$, and not $E_S = E_{\{S0\}} \cos(2\pi Q/2e)$. This is in full similarity to how the Josephson energy of a symmetric SQUID is $E_J = E_{\{J0\}} |\cos(\pi\Phi / \Phi_0)|$. On the other hand, if the two junctions were strongly asymmetric, I would expect $E_S = E_{\{S1\}} + E_{\{S2\}} \cos(2\pi Q/2e)$ with small oscillation amplitude, $E_{\{S2\}} \ll E_{\{S1\}}$. This does not match Eq. (4) either.

– The authors expect that the phase slip energy in their device is $E_{\{S0\}} / h = 15$ GHz. In theory, this should correspond to the critical voltage $V_C = \pi E_{\{S0\}} / e = 200\mu V$. At the same time, the observed Coulomb blockade voltage is around $2.5\mu V$. What is the reason for such a huge gap between the observed and expected values?

– Currently, the main novel element of the paper – the effect of Aharonov-Casher interference on dual Shapiro steps – is confined to a single figure showing processed data [Fig. 3]. From this figure alone, it is difficult to assess the degree of tunability of dual Shapiro steps by gate voltage. Could the authors provide examples of $I(V)$ traces at different gate voltages, please (e.g., in the supplementary materials)? I think it would be very helpful if they could show $I(V)$ both at the maximum and at the minimum of the interference pattern. These data can, in principle, be used to assess how asymmetric the junctions in the authors' device are.

– Even when there are no microwaves [left panel of Fig. 2], $I(V)$ has a structure with extra peaks, in addition to the main Coulomb-blockade peak. What is the origin of these extra peaks?

– The authors repeatedly say that the Bloch transistor is a non-dissipative element. I don't understand this statement. The authors' device has two normal state resistors. The fact that the device is dissipative is clearly exemplified by the left panel of Fig. 2 in which the dissipated power $P = I V$ is non-vanishing.

Minor feedback:

– Parameter V_C is not defined when it is first used [see the text after Eq. (3)].

– It is not clear from the text what thermal current δI_T means. While the authors define it in a related publication [Nature Communications 15, 9326 (2024)], but a clarification in the present manuscript would be helpful.

– Eq. (1) in the supplement introduces the resistance R . Please define R . Is this the total resistance connected in series with the junctions?

– In the beginning of Sec. III, as well as right before Fig. 3, the authors say that the critical voltage is around $10 \mu V$. At the same time, it is clear from Fig. 2 that $V_C \approx 2.5 \mu V$. Which value is the correct one?

Typos and stylistic remarks:

– The abbreviation ISS is only used where it is introduced. Is it needed?

– In the text, the authors introduce the critical Coulomb blockade voltage V_C . On the other hand, in the figure they use

V_C*. Is V_C* the same thing as V_C?

– In Fig. 2 caption, the authors use I_A*. Is this a typo, and should read I_C*?

– The authors say “It is not easy to fabricate JJ smaller than $40 \times 40 \mu\text{m}^2$ in a controllable way.” This must be a typo, and they actually mean $40 \times 40 \text{ nanometer}^2$.

Reviewer #4

(Remarks to the Author)

Reviewer #5

(Remarks to the Author)

Version 1:

Reviewer comments:

Reviewer #1

(Remarks to the Author)

In my review of the original manuscript, I noted that the work by Antov et al. has the potential to generate significant interest within the broader physics community. However, I also highlighted several important weaknesses, including:

- Unclear main achievement
- Significant overlap with previous work
- Ambiguity surrounding the biasing scheme
- Incomplete literature review and lack of comparisons with prior superconducting transistors
- General style and presentation issues

Having reviewed this new version, I can see that the authors have made significant improvements to their manuscript. For instance, they completely rewrote the abstract and introduction, which now clarify their main claim and provide a better overview of the existing literature. Additionally, they have greatly expanded the Supplementary Information, making the experimental techniques more understandable.

Unfortunately, I still have concerns regarding the general style and logical flow of the article. For example, the name "Aharonov-Casher" is misspelled several times throughout the paper, despite being a central mechanism upon which Antov et al.'s work is built. This is just one example among many in the manuscript. My intention is not to hinder the publication of this paper; rather, I strongly encourage the authors to utilize available proofreading tools to ensure that their manuscript meets the standards of Nature Communications.

Reviewer #3

(Remarks to the Author)

I'd like to thank the authors for incorporating my feedback. After the authors restructured the manuscript to a physics-based narrative – as opposed to the new device-based narrative – the paper reads much better in my opinion. I also appreciate the effort in making the explanations clearer, and the estimates of the Bloch transistor parameters (such as the apparent critical voltage and current) more consistent. Lastly, in my view, it works to the scientific benefit of the paper that the authors are upfront about the relative magnitude of the gating effect (“The gating effect is weakly pronounced in the direct I - V curve,...”, “The amplitude of dV/dI modulation can be as large as 14%”).

While the quality of the paper improved significantly, I still stand by my opinion that the results represent a relatively minor advancement over the authors' previous works, and as such, belong better in a more specialized journal, e.g., Communications Physics. The data presented in the paper will surely be of interest to physicists very closely following the progress in areas of “dual” superconducting electronics and superconductor-insulator transition in Josephson junctions. However, I don't think it represents an important advance of significance to a broad physics community.

Comments related to the new version of the manuscript:

(i) I don't understand the authors' point about “the new mechanism of phase locking.” Specifically, they claim that the mechanism of phase locking is different in a two-junction system as compared to a single-junction one: in a two-junction system, the phase locking is the result of oscillating charge on the island, whereas in a single-channel system the phase

locking is the result of the oscillating current. In my view, presenting these as two distinct mechanisms is an overstatement. In both cases, the phase locking can be simply attributed to the oscillations of the offset charge across the junction. The offset charge itself can be induced in different ways but the fundamental mechanism is the same. Indeed, consider the argument of the Bessel function in the case of the current bias [Eq. (4) of Nat. Comm. 15, 9326 (2024)]; what it contains is I_{ac} / f which is nothing but the offset charge.

(ii) The choice of “shifting the curves for clarity” in Fig. 3(b) is a poor one in my opinion. To start with, it’s not clear which of the curves is not shifted. What’s more important is that, with the shifts, it’s very hard to get an idea of how large the gate-voltage modulation of dV/dI is compared to the overall magnitude of dV/dI . The text mentions 14% but it’s hard to see from the plots. As a reader, I’d rather prefer to see individual traces as individual panels, even if that would make the figure more bulky.

(iii) Please define R after Eq. (3).

(iv) In the sentence “The MW control of the BT with I_{ac} and f comes from...” the parameter I_{ac} is not defined in the current version of the paper. I presume the authors ment δQ_g .

Reviewer #4

(Remarks to the Author)

The images or other third party material in this Peer Review File are included in the article’s Creative Commons license, unless indicated otherwise in a credit line to the material. If material is not included in the article’s Creative Commons license and your intended use is not permitted by statutory regulation or exceeds the permitted use, you will need to obtain permission directly from the copyright holder.

Dear Editor of *Nature Communications*,

Please find the point-by-point answers (red ink) to referee questions below.

Sincerely Yours,

V Antonov

on behalf of all co-authors.

Reviewer #1 (Remarks to the Author):

Comments on the Manuscript "Bloch Transistor for Cryogenic Quantum Electronics"

This manuscript presents a superconducting device comprising two Josephson junctions in series, which form a small superconducting island and are embedded in a high-impedance circuit that ensures charge quantization and isolation from external noise. This device is referred to as a Bloch transistor. While the topic has potential interest for quantum electronics and metrology, I believe that the current version of the manuscript does not meet the standards of Nature Communications and cannot be considered for publication for the following reasons:

1. Unclear Main Achievement

The main contribution of the paper is not clearly defined. Is the primary claim the Bloch transistor itself? The electrical control of dual Shapiro steps? The potential application for controlling quantum devices? The introduction and title emphasize the potential application of the device for controlling quantum systems; however, there is no data presented that demonstrates how the Bloch transistor might be superior to existing control methods. Can the authors provide evidence that it is more stable, more precise, or less noisy?

In my opinion, the authors need to choose whether they want to publish a physics paper or focus on presenting a device with potential applications. Currently, the title and the introduction suggest an emphasis on applications, yet the performance of the device is not addressed or compared to existing alternatives. If the authors intend to explore the physics of the Bloch transistor, they should significantly rewrite the paper and clearly explain what makes the physics of this device novel and distinct.

The Bloch Transistor is a conceptually new quantum system combining two phenomena: the Dual Shapiro Steps and the Aharonov-Casher effect. The synergy of two effects opens a new dimension in fundamental physics, similar to the move from the JJ to the dc SQUID. To respond to the referee criticisms we substantially modified the text to clarify the motivation of the research:

We report on observation of two coherent quantum phenomena, the current quantization in the Josephson Junction (JJ) and the Aharonov-Casher effect. The synergy of two effects is the phase locking of the JJ with the fluctuating charge engaged by the microwaves (the current quantization), and a control of the phase locking phenomenon with the static charge of the gate electrode (the Aharonov-Casher effect). Conceptually, the system has the functionality of the Bloch Transistor: it can deliver gate-controlled quantized non-dissipative current to the quantum circuit. We changed the title to reflect the motivation of the research: “The microwave phase locking in Bloch Transistor”.

We agree with the referee, that the BT is not elaborated down to the application. Also it is premature to discuss the superiority of BT over the devices and methods of the traditional cryogenic electronics as the accuracy of the current quantization, the depth of gate modulation and the yield of the devices are far from the requirement of the sensible application.

2. Significant Overlap with Previous Work

The manuscript exhibits substantial overlap with previously published work by the authors, particularly:

- Nature 2022, s41586-022-04947-z
- Nature Communications 2024, s41467-024-53600-y

The only significant modification seems to be the transition from a single junction to a series of two (i.e., the transistor configuration), which does not, in itself, constitute a sufficiently novel contribution.

Transition from a single JJ to a series of two is substantial, as the device starts to operate on the new physical principle of phase locking for the current quantization. Instead of *ac* current modulation in the single JJ, the charge on the island is modulated in BT. This allows a new functionality, the gate control of the quantized non-dissipative current (Aharonov-Casher effect). The concept was theoretically predicted by Erdmanis&Nazarov (Erdmanis and Y.

Nazarov, *Physical Review B* 106, 10.1103/physrevb.106.235406 (2022)).

The transition is similar to the case of a *dc* SQUID, where new physics emerge when the loop with two JJs replaces an individual JJ.

The only data that could be interpreted as demonstrating a new feature is presented in Figure 3, which relates to the gating of the superconducting island. However, this figure is confusing: it presents only differential conductance, rather than the actual current, which is more relevant for transistor operation. Additionally, the authors mention that "the current plateaus dissolve and reappear with the MW amplitude," but the influence of the gate voltage on the main features (as shown in Figure 2) is not adequately presented. The discussion primarily focuses on current metrology, similar to the authors' previous works.

We disagree with the referee on these matters. The new findings are in Figures 1, 2 and 3. Figure 1 demonstrates the new concept of the BT, Figure 2 demonstrates the current quantization due to the new mechanism of the phase locking, and Figure 3 shows the modulation of the quantized current with gate voltage, which has not been demonstrated before. In the revised manuscript, we also add details of a new model of the phase locking in BT. The gate modulation presents a change in the quantised current plateaus flatness. In the current experiment this modulation is limited. At the same time, the differential resistance is the natural measure of flatness, which can clearly demonstrate the effect. There are a few factors which limit the modulation: the poisoning of the BT with the quasiparticles (parity effect), which halves the depth of modulation; the difference in the phase slip energies of the two JJ; the thermal noise I_T . Certainly, further development is needed to increase the modulation depth.

3. Ambiguity Surrounding the Biasing Scheme

The measurement setup is not sufficiently described in the main text or the supplementary material. This omission is particularly problematic regarding the IV curves shown in Figure 2. It is likely that these curves include a resistance correction to display "back-bending" features, which gives the misleading impression that multiple current values correspond to the same voltage—a concerning trait for a transistor or current source.

The measurement scheme is a mixture of a voltage and a current bias circuit. We use a symmetric four-point measurement scheme where voltage and current are measured independently of the bias voltage V_b . The measurement scheme allows us to record the back-bending directly. The data shown in Fig. 2 do not have any correction. We add a corresponding explanation to the main text and a reference to the principal circuit of the differential amplifier used:

“For *dc* measurement we use four-point methods with the symmetric differential amplifier. The measurement is a mix of the current and voltage bias scheme, where both, the voltage and current, are probed independently of the bias voltage V_b . The setup enables the measurement of *I-V* curves with back-bending, as shown in Fig. 2(a). The principal scheme of the amplifier is given in the Supplementary note 2.”

In reality, the uncorrected data would likely show a single current for each applied bias voltage, with quantized current values as the bias voltage varies. In this context, the role of the gate voltage is unclear.

The curve shown in Fig. 2 has original unprocessed data.

The authors should clarify the operational advantages of using the gate voltage in comparison to the bias voltage and explain how gating enhances or differentiates this device from their previous designs.

For the optimal operation of BT, there is a set of parameters- the bias and gate voltages, and the amplitude of the *ac* signal. The number of controls provides the freedom to set the optimal operating point. The gate control has an advantage over the others in operation speed, independence from history, and flexibility of operation.

To the best of our knowledge, no device so far has gate control of quantized non-dissipative current.

****Two related notes:****

- Presenting the uncorrected IV data would be beneficial in evaluating the actual extent of the current plateaus, specifically how many data points contribute to each step. This is crucial for assessing the precision required for voltage control in practical applications.

The number of the experimental points forming the plateau with $n = 1$ on Fig. 2(b) is 66. The width of the plateau is $\sim 1 \mu\text{V}$.

- What do the authors mean by "One can tune the BT to different n by varying V_b " (Figure 2 caption)? The current form of this figure indicates that multiple currents can be obtained for the same V_b . An uncorrected IV curve would likely demonstrate more clearly that different n values can be accessed by varying V_b .

One can move between the plateaus with different n by changing V_b . The effect is due to non-linear relation between the bias voltage, V_b , and the voltage across BT, V . The result depends on the history of biasing, however. This should be taken into account when using the voltage bias control. For example one can move from $n = 1$ to $n = 2$ by increasing V_b continuously from small to high bias voltage.

The $I - V$ curves presented in the manuscript use uncorrected data.

4. Incomplete Literature Review and Missing Comparisons with Prior Superconducting Transistors

The manuscript does not adequately acknowledge a significant body of literature related to superconducting transistors and other gate-modulated devices. Among the relevant references that have been omitted are:

- PRL 92, 066802 (2004)
- PRL 95, 206807 (2005)
- Nature Physics 12, 825–829 (2016)
- PRL 98, 216802 (2007)
- PRB 99, 220504 (2019)
- PRResearch 4, 013038 (2022) and the references therein.

A meaningful comparison with at least the most technically similar devices is essential. Furthermore, to justify the significance of this work, the manuscript should clearly articulate how the presented Bloch transistor offers tangible improvements over these earlier proposals and demonstrations, which is currently lacking.

We appreciate the referee's suggestion and have added the references to PRLs and PRB papers to the manuscript. Indeed, most of them are concerned regime where $E_C \sim E_J$, which is the prerequisite of the CQPS operation. The focus of the publications was mainly on the parity issue and energy spectroscopy of the Cooper Pair Transistor. We have included the references with the text:

“Interest to this regime concerned mainly the parity issue and energy spectroscopy of the Cooper Pair Transistor [10-12]. Less attention was given to the CQPS. The study of BT operation would shed light on the fundamentals of the CQPS phenomenon and its implementation to quantum devices with new functionalities.”

There is an error in the citation of Nature Physics 12, 825–829 (2016). We did not manage to identify the source of the article.

The last reference PRResearch 4, 013038 (2022) is the theory suggesting the topological quantum circuit. Despite the high quality of the work, so far, there is no experimental confirmation of the operation of the circuit. We hesitate to add a reference to this work.

5. General Style and Presentation Issues

The manuscript lacks clarity and the main claim is not properly defined. The structure could be significantly improved to better guide the reader through the novelty of the device and present the experimental results in a more intuitive and coherent manner.

We put efforts into improving the clarity and focus of the manuscript.

Conclusion

In its current form, this work appears to be a largely incremental advance over the authors' previous studies. The lack of a clear conceptual motivation, combined with insufficient data presentation and the absence of comparison with existing literature, makes it difficult to justify publication. Significant revisions, both in terms of scientific content and clarity, would be required to bring the manuscript to Nature Communications standard.

The original submission of the manuscript was to Nature Electronics with focus on principle of operation and functionality. The manuscript was transferred to Nat Com without modification, which unavoidably attracted fair criticisms from referees. In the current version, we expand the explanation of the underlying physics, add the theoretical model, references to the related works and respond to referee questions.

We report significant new physics when a single JJ in a regime of CQPS is replaced with two close JJs. The mechanism of the phase locking by the MW signal then changes from the oscillating current to charge. This allows not only to have the current quantization in a system, but also to control this quantization through the Aharonov-Cahier effect. The BT, in some sense, is dual to the dc SQUID. We believe that nobody would argue that dc SQUID is an incremental step from the single JJ.

General Remarks on the "Bloch Transistor for Cryogenic Quantum Electronics"

1. Quantum Dissipative Nature of the Device: The authors describe the device as purely quantum and non-dissipative in the abstract. However, they discuss operating the junction in the "Bloch regime," which actually involves a dissipative environment.

Indeed, the screening circuit of BT has the dissipative Pd resistors. However tunnelling of the fluxes across the JJ, and current of the Cooper pairs are non-dissipative. We can safely state that between the inductors denoted by L_1 the circuit is non-dissipating.

The only data that could be interpreted as demonstrating a new feature is presented in Figure 3, which relates to the gating of the superconducting island. However, this figure is confusing: it presents only differential conductance, rather than the actual current, which is more relevant for transistor operation. Additionally, the authors mention that "the current plateaus dissolve and reappear with the MW amplitude," but the influence of the gate voltage on the main features (as shown in Figure 2) is not adequately presented. The discussion primarily focuses on current metrology, similar to the authors' previous works.

The phase-locking mechanism in BT is different from a single JJ, as we explained in the manuscript. New mechanism of phase locking allows a gating effect, which is impossible in single JJ. At the same time, the appearance of the current quantization at the fixed gate voltage is similar to that reported before.

We use differential resistance in Fig. 3 for clarity. We responded to this question above.

3. Ambiguity Surrounding the Biasing Scheme

The measurement setup is not sufficiently described in the main text or the supplementary material. This omission is particularly problematic regarding the IV curves shown in Figure 2. It is likely that these curves include a resistance correction to display "back-bending" features, which gives the misleading impression that multiple current values correspond to the same voltage—a concerning trait for a transistor or current source.

In reality, the uncorrected data would likely show a single current for each applied bias voltage, with quantized current values as the bias voltage varies. In this context, the role of the gate voltage is

The authors should clarify the operational advantages of using the gate voltage in comparison to the bias voltage and explain how gating enhances or differentiates this device from their previous designs.

****Two related notes:****

- Presenting the uncorrected IV data would be beneficial in evaluating the actual extent of the current plateaus, specifically how many data points contribute to each step. This is crucial for

assessing the precision required for voltage control in practical applications.

- What do the authors mean by "One can tune the BT to different n by varying V_b " (Figure 2 caption)? The current form of this figure indicates that multiple currents can be obtained for the same V_b . An uncorrected IV curve would likely demonstrate more clearly that different n values can be accessed by varying V_b .

4. Incomplete Literature Review and Missing Comparisons with Prior Superconducting Transistors

The manuscript does not adequately acknowledge a significant body of literature related to superconducting transistors and other gate-modulated devices. Among the relevant references that have been omitted are:

- PRL 92, 066802 (2004)
- PRL 95, 206807 (2005)
- Nature Physics 12, 825–829 (2016)
- PRL 98, 216802 (2007)
- PRB 99, 220504 (2019)
- PRResearch 4, 013038 (2022) and the references therein.

A meaningful comparison with at least the most technically similar devices is essential. Furthermore, to justify the significance of this work, the manuscript should clearly articulate how the presented Bloch transistor offers tangible improvements over these earlier proposals and demonstrations, which is currently lacking.

5. General Style and Presentation Issues

The manuscript lacks clarity, and the main claim is not properly defined. The structure could be significantly improved to better guide the reader through the novelty of the device and present the experimental results in a more intuitive and coherent manner.

Duplicated questions 3-5. They have been answered above.

Conclusion

In its current form, this work appears to be a largely incremental advance over the authors' previous studies. The lack of a clear conceptual motivation, combined with insufficient data presentation and the absence of comparison with existing literature, makes it difficult to justify publication. Significant revisions, both in terms of scientific content and clarity, would be required to bring the manuscript to Nature Communications standard.

Duplication, responded above.

2. Junction Parameters: The junction parameters appear to be quite high, and they seem to shift unexpectedly at various points throughout the paper. For example, using the provided values for E_J and E_C to calculate the plasma frequency yields a frequency around 200 GHz, which is significantly higher than the aluminum gap. The paper would benefit from a clearer explanation or justification of these parameter choices.

The choice of JJ parameters is dictated by condition $E_C \sim E_J$. For such JJs, the plasma frequency exceeds the gap of Al, and it cannot be reached. The phase slip energy is important for the CQPS operation. It is smaller than the superconducting gap.

3. Calculation Accuracy: Several calculations in the paper are not performed with sufficient care, and noticeable errors are present. For instance, based on the given E_J , the critical current I_C of a single Josephson junction should be approximately 180 nA. However, the paper mentions a value of 100 nA, despite initially defining I_C as the critical current of one junction. Additionally, when calculating the apparent current I_C^* , the authors use the formula $I_C^* = (\pi E_J) / (16 E_C) * I_C$. Plugging in the values results in approximately 64 nA, while the paper states 14 nA, which is unexpected.

Starting from the normal resistance of the JJ, 1.7 kOhm, and superconducting gap of Al, 202 μ V, one can estimate the critical current $I_C \sim 186$ nA. Correspondingly, $E_J/h = 92.5$ GHz. We carried out this calculation on page 2 to estimate the phase slip energy E_{S0} and the apparent critical voltage V_C^* . The latter is 3.5 μ V, which is close to the experimental value of 2.5 μ V.

There is an issue with the experimental value I_C of the individual JJ: it is ~ 100 nA, which is smaller than estimated, ~ 186 nA. It may be due to the back action of the measurement system. We use experimental I_C to estimate apparent critical current $I_C^* \sim 17$ nA on page 3 (top of second column). It is close to the experimental value of 14 nA.

4. Missing Error Bars and Unexplained Behavior in Fig. 4: The plot in Fig. 4 (the one on the right) lacks error bars, making it difficult to assess the quality of the fit. Furthermore, the differential conductance (dV/dI) does not approach zero at the origin of the applied AC current, yet the paper does not describe or explain this inconsistency.

Standard deviation of the resistance is ~ 50 Ohm, the error bars are not resolved in the graph. The effect of rectifying white noise by the circuit is evident at low MW power. It is seen as non-zero step width even at zero MW power.

5. Insufficient proofreading of the manuscript: for example, the current title of the supplementary material is “Quantized Current Steps Due to the Synchronization of Microwaves with the Bloch Oscillations in Small Josephson Junctions.”

Proofreading has been done. The supplement title is amended.

Reviewer #2 (Remarks to the Author):

Review on the manuscript: Antonov et al., “Bloch transistor for cryogenic quantum electronics”

Bloch transistor (BT, also: “Bloch electrometer”, “Cooper pair transistor”, etc.) - two small, closely spaced Josephson junctions, in which the charge on the island between the junctions can be varied by an external gate voltage – has been extensively studied since late 80th; theory and experiments published in a few tens of papers. The charge dynamics in this circuit reflects the interplay of Josephson and Coulomb effects, which, in the case of the dominating Josephson energy and low-bias operation, can be reduced to that of a single small Josephson junction having a gate-controlled Josephson coupling $E_J = E_J(Q)$, where Q is a gate-induced charge of the island. A current I through a small Josephson junction is related to the so-called Bloch oscillations which are oscillations of voltage across the junction with a period proportional to

$2e/I$. It has been a long-standing but challenging idea to lock the Bloch oscillations to a microwave drive, and so to generate steps of constant current, dual to Shapiro steps in the Josephson voltage standard. For very dissimilar Bloch and the drive frequencies, the resonant mixing of Bloch oscillations with the probe signal was implemented in a reflectometry setup published in Nguyen et al., PRL 99, 187005 (2007). A remarkable advantage of the experiment under review is that such an effect is reported now for matching frequencies and registered in form of DC current steps (so-called dual Shapiro steps) in the IV-curve of a Bloch transistor embedded in a high-impedance biasing circuitry.

The reported achievements mark a recent progress in the field after publishing Refs. [8-10] as well as other important papers during the past few years. Correspondingly, this new study can be understood at best in the tight context of the preceding works, but it does not represent a crucial “game changer” in the field. The data are definitely of substantial interest for scientists involved in the studies of small Josephson junctions. However, at the present form the manuscript should not be published and it needs, in my opinion, to be conceptually reworked.

Below I will list a few points of major and minor criticism, in the hope it could help to produce a high-quality paper of profound interest.

1. The title of the manuscript ending by “... for cryogenic electronics” implies a high degree of universality of the message, but up to now there has been only a few applications, envisaged solely in metrology, related to the fundamental current-to-frequency relation, $I = 2e \times f$, which are in focus of the present study.

We have amended the title to reflect the physics behind the experiment: Phase locking of the Bloch oscillations by the MW in BT. The mechanism of phase locking in BT is different from that in a single JJ, as we explained above. The new mechanism allows the gating effect, which is impossible in a single JJ. It is a “game changer” in the field. Application of the BT as the current standard and a source of non-dissipative quantized current are two straightforward applications.

2. In the Introduction, there is a very scarce review of previous works related to Bloch transistor, despite the decades-long and extensive scientific context behind. The cited few works are

mostly related to superconducting microconstrictions or single tunnel junctions and the way to interpret them in terms of coherent phase slips, but not to BT.

We have added an overview and references to works dealing with superconducting of different functionalities. Some of them indeed are called the Bloch transistor. In our work we refer to the Bloch transistor as a device with new functionalities: the Bloch oscillations are phase-locked with the MW, producing a quantized current. The latter is controlled by the electrostatic gate.

3. In the Introduction, a need for low-noise current sources for quantum devices is used as motivation for the work. On the other hand, it appears obvious (and is repeated several times in the manuskript) that the transistor itself requires four external signals for operation, which would certainly be the same way not free of noise as any other. The effect of noise is known to increase the linewidth of Bloch oscillations leading to fluctuations of the nominally quantised DC current component. Moreover, the output signal across the dual Shapiro steps will necessarily involve, along with the DC term, also a long set of harmonics as well as the trace of the driving signal itself.

The current is the flow of the individual electrons by nature. In case of the quantized current this flow has periodicity of driving frequency, $I=2efn$. No other harmonics are present. For the time interval much larger than $1/f$, one has a *dc* current in the circuit.

Another point: proper operation of BT was shown in the paper to happen in a very narrow, sub- μ V bias range. This leads to extremely high sensitivity of BT to the back-action processes in the possible load. As a quantum circuit, the load will be typically controlled and manipulated by GHz signals thus producing inevitably strong disturbances to BT. Neither the accuracy nor the current level of the dual Shapiro steps presented in the paper are superior to those previously reported for microconstrictions/single junctions in Refs. [8-10]. The argument is then open why using BT instead of much simpler single junction circuits. To summarize, I would not expect any impact of Bloch transistors as a current source for quantum circuits supporting the statements declared.

We agree with the referee that the back action of the load circuit is an issue. It is not explored

in our work. The advantage of the BT over the single JJ is the gate control of quantization.

On the other hand, I would like to stress once again the scientific interest and high relevance of the results obtained. In particular it is so, because of an intriguing, in fact, parametric - due to the dependence $ES(C_g \times V_g)$ - mechanism of coupling the microwave drive and the Bloch oscillations in the setup under study.

We appreciate the referee's comment. The new phase locking mechanism is demonstrated in our work.

4. Starting at the introduction and following in the next section, the phenomena are treated by focussing on the coherent flux tunneling and Aharonov-Casher effect, which is a usual treatment of superconducting CQPS microconstrictions. This viewpoint is dual to that describing the coherent Cooper pairs transport in single charge transistors and by far not intuitive, nor insightful for a wide-scope scientific journal. Just to note it briefly, for small Josephson junctions there has been essentially similar, but more direct formalism based on charge-phase Hamiltonian adopted by Averin, Zorin, and Likharev in Sov. Phys. JETP, 61 (2), 407 (1985) and further developed towards Bloch transistor in Zorin, PRL, 76, 4408 (1996).

Whatever approach is used, it should be self-explaining in respect to the phase locking mechanism. In BT the mechanism is expected to be different from that of a single junction (or QPS junction), because it does not involve an RF driving signal injected in parallel to the DC bias, but rather the parametric gate-modulation of the Josephson coupling strength (cf. discussion around Eq. 4 in the manuskript). Under this condition, the equation (3) developed for a single junction may not be applied directly to BT and needs to be replaced by a more dedicated expression.

We agree with the referee that the Aharonov-Casher effect is not easily understandable for a broad readership. To address this problem, we have developed a simple model based on the formalism developed by Averin, Zorin, and Likharev in Sov. Phys. JETP, 61 (2), 407 (1985), which is, indeed, very convenient. We hope that this model is easily understandable. The phase locking mechanism is the same as in a Josephson junction because our model is mathematically

equivalent to that of a Josephson junction. With this model we re-derive Eq. (3) assuming that the AC signal is applied to the gate electrode. In this way, we show that the argument of the Bessel functions should be replaced, $I_{ac}/(2ef) \rightarrow \delta Q_g/2e$, where δQ_g is the amplitude of AC modulation of the gate charge.

Full analysis and calculations are given in Supplementary Note 3. Corresponding amendments are made in the main manuscript.

In this respect, it is also not clear (neither defined explicitly in the manuscript) which quantity is denoted by I_{ac} ? Due to the strong Josephson coupling, the individual junctions in BT can not be considered as discrete components experiencing individual gate injection, but even if they could, the sign and value of two injected currents would be two deviating parameters instead of one I_{ac} .

The referee is correct. In our case, the mechanism of phase locking is performed through the induced charge at the island, δQ_g , rather than current I_{ac} , like in a single JJ.

To my understanding, developing a model of dual Shapiro steps in BT based on the parametric driving effect would be of significant importance for understanding the experimental findings reported.

Finally, the logical structure of the manuscript would benefit of a more concise model description without scattering the details across sections I, II, III as now.

The referee is correct. We amended the manuscript accordingly to reflect the focus on the new mechanism of phase locking.

A few minor points are listed below:

1. In Fig. 1 the bias voltage (and the biasing layout generally) is drawn one-sided and open-ended. This deviates much from the usual (and de-facto implemented on chip) circuit design of BT and, strictly speaking, is unphysical.

The figure is amended.

2. In Fig. 1 the quantities L_1 and L_2 are mentioned but not labeled.

L_1 and L_2 notations are added to Fig. 1.

3. Page 2, left: «Inverse Shapiro Steps» is not widely called so in literature. The most used term is “dual Shapiro steps”.

We have change to Dual Shapiro steps

4. Bottom paragraph of Page 2, left: Passive circuit components usually constitute the EM environment, but they do not belong directly to BT.

We cut this statement.

5. It is not clarified in the text in which way the tunneling energy of the fluxon E_S is related to the junction current. Surely, one can learn more from Ref. [8], but that would render the main text being not self-explaining. See also above the comment to the model.

The E_{S0} is related to I_C through E_J , which can be rendered from the text around Eqn. 2.

6. Equation (2) is neither derived here, nor referred to a literature source. From Ref. [13], the expression for the plasma frequency is indeed available, but not the expression for E_S . From the text flow, it is not quite clear, either the quantity E_{S0} is related to a single junction or to BT (“In the BT under study $E_{S0}/h \sim 15$ GHz”)

We have added a reference for Eq. 2 and expanded the discussion related to E_{S0} . The E_{S0} is calculated for a single JJ.

7. Page2, right: “Finally, this parameter determines the accuracy of the current quantization and the operating temperature.” This statement is not clear and should be explained. It is also not obvious how the temperature depends on E_{S0} .

The temperature affects V_C^* through thermal noise δI_T . Further, the flatness of the quantized plateau depends on the ratio of eV_C^* to $k_B T$.

8. “The BT delivers quantized current when it is voltage-biased,..” This statement, if assumed to be valid for a gate-voltage driven BT, should be justified or referred to literature (theory?). Furthermore, it apparently contradicts to the requirement of high-impedance biasing circuit for observation of Bloch oscillations. In case of pure voltage bias, no voltage oscillations can just appear across the double junction.

We have amended this statement at the bottom of first column on page 4:

“The BT can potentially deliver non-dissipative quantized current to the quantum circuit with high accuracy”.

The current can be delivered to the circuit embedded in a scheme between two L_{1S} .

9. The parameter V_C is used before it is defined in the text.

We define V_C^* in Equation 3.

10. Equation (4) is neither derived, nor referred to a literature source.

In the current version, this equation is (5). We add reference [2] describing the derivation of the equation. Also, an explanation is given in Supplementary Note 3.

11. Fig.2, caption (also in the text): “One can tune the BT to different n by varying V_b .” From Fig. 2 it is not so obvious that one can reach, say, the step at $n = 0$ when starting from other current steps and varying the bias voltage only. As shown in Fig. 1 and in supplementary information, the IV-curve was measured in a four-point layout utilising a current bias and a voltage measurement procedure.

We repeat the explanation is given above:

We use a symmetric four-points measurement scheme with voltage and current are measured independently of bias voltage. It is a mixture of voltage and current bias circuits. Such a measurement scheme allows measuring back-bending directly. The confusion came from the

wrong x- axis label in Fig. 2: it should be V instead of V_b . We amended the error in Fig. 2, and added text to the manuscript:

“For *dc* measurement we use four-point methods with the symmetric differential amplifier. The measurement is a mix of the current and voltage bias schemes, where both the voltage and current are probed independently of the bias voltage V_b . The setup allows for measuring I - V curves with back-bending, as shown in Fig. 2(a). The principal scheme of the amplifier is given in the Supplementary Note 2.”

Bias control depends on the previous history. We have added text:

“Such a control is possible because in our measurement scheme the relation between the voltage applied to the circuit, V_b , and the voltage measured across the BT itself, V , is not linear. The current-voltage relationship does not have one-to-one correspondence, however, and depends on the history.”

12. The numeric values of some quantities seem not to be quite consistent. For example, cf. “capable of delivering a quantized current up to 10 nA.” at the end of the Introduction, 14 nA in conclusions and “quantization is present between 3 GHz and 20 GHz” in Section III. On the contrary, the resistive (Landau-Zener) branch in Fig.2 sets on at about ~ 5 nA which is even lower. The resistance scale of ~ 1 kOhm in Fig. 3 is much lower than stated: “The differential resistance of the quantized plateaus is below the “blockade” resistance of 20 k Ω ”. Similar deviations can be found in respect to the bias voltage estimates up to 10 μ V, while the plotted values are all much lower. The same for V_g .

We have corrected some of the values, some of them stay as they are:

1. $V_C^* = 2.5 \mu$ V, which is in Fig. 2, corrected
2. Quantized current up to 6.6 nA, corrected
3. In the model sample the frequency range explored spans from 6.7GHz to 10.4 GHz. 4. The Zener current in Fig. 2(a) is ~ 14 nA.
5. The differential resistance at the plateaus of current quantization is smaller than 20 kOhm at the current blockade region of the I-V curve without MW.
6. We corrected the text: the blockade voltage is $\pm 2.5 \mu$ V.
7. The gate voltage was varied within the range ± 2.5 mV, see Fig. 3(b).

13. Page 3, right: The explanation of the gate voltage dependence related to Fig.3 in terms of fluxes and Aharonov-Casher interference is not directly insightful for interpreting the differential resistance of RF-driven BT, especially under the large gate signals sweeping over several periods, $Q > n \times 2e$.

We have amended the explanation of the gating effect and added the full model in the Supplementary note 3

14. For Fig. 4 neither bias nor dc gate voltage values are indicated (even if maybe not so substantial for the pattern).

We believe it is natural to show differential resistance in coordinates of the quantized current $I_{dc}/2ef$ and the normalized oscillating charge at the island $\delta Q_g/2e$

15. Since the microwave power values are provided in the normalized form as I_{ac} , it is difficult to conclude how strong the signal was in terms of $Q/2e$ and either there was an effect of mutual synchronisation of two Bloch oscillations: the one along the bias path and the other induced by the gate to the island as a box (similar to Nguyen et al. cited above).

We change the axis from $I_{ac}/2ef$ to $\delta Q_g/2e$ to reflect the phase locking mechanism.

Finally, I would like to encourage the writing authors to further optimize the text flow towards easier understanding of physics behind the experiment.

We revised the manuscript's structure and text to address the referee's suggestions.

Reviewer #3 (Remarks to the Author):

When a Josephson junction is embedded into a high-impedance environment, its physics becomes dominated by effects of electric charge discreteness. One such effect is Coulomb blockade. The blockade impedes the flow of supercurrent, and makes the junction display

insulating (as opposed to superconducting) signatures. Another effect is Bloch oscillations. These are the oscillations of voltage across the junction biased by direct current I . The Bloch oscillations occur with a frequency $f = I / (2e)$ that only depends on the value of the current and unit of charge $2e$. It is the latter effect that is the cornerstone of the authors' work.

One can also consider the phase slip model, where voltage is induced due to periodic flux tunnelling across the JJ.

Specifically, the authors introduce a device called a Bloch transistor. In this device, two Josephson junctions are connected in series, and an island between them is capacitively coupled to a gate electrode. The high-impedance environment is provided by on-chip resistors and inductors. The two-junction structure allows one to control not only the frequency, but also the amplitude of Bloch oscillations, through the Aharonov-Casher effect (i.e., by varying the gate voltage).

The authors propose to use the Bloch transistor as a source of quantized current. This application can be achieved by synchronizing Bloch oscillations with external microwave radiation. The microwave frequency f sets the quantized values of the current, $I = n * 2ef$.

To verify the feasibility of this application, the authors measure the current-voltage characteristics $I(V)$ of the device in the presence of microwaves. They observe that $I(V)$ develops a set of quantized steps, at the expected values of I (the so-called dual Shapiro steps). They also show that the sharpness of the steps – and thus the precision of the quantization – can be controlled by varying the gate voltage; this observation is in qualitative agreement with the Aharonov-Casher effect.

Overall, the authors' results make sense to me. The data convincingly demonstrates the dual Shapiro steps in the Bloch transistor; it also illustrates the possibility of controlling the parameters of the steps using gate voltage, and frequency and amplitude of microwaves. Unfortunately though, I do not get an impression that the results represent a significant advancement over the previous works. I also have questions and concerns regarding the authors'

analysis. Therefore, I cannot recommend publishing the article in Nature Communications at this stage. I elaborate on the reasons for my assessment below.

To begin with, let me note that both the dual Shapiro steps and the Aharonov-Casher interference were previously demonstrated. In the former case, this includes a series of works by the same authors on superconducting nanowires [Nature 608, 45 (2022)] and Josephson junctions [Nature Communications 15, 9326 (2024), Appl. Phys. Lett. 125, 122602 (2024)], as well as works by other authors [Nature Physics 19, 851(2023); Nature Communications 15, 8726 (2024)]. Aharonov-Casher interference – i.e., the interference between the coherent phase slips – was also observed in a number of contexts, including that of the Josephson junction arrays [Phys. Rev. B 85, 094503 (2012), Phys. Rev. B 85, 024521 (2012)] and disordered superconductors [Nature Physics 14, 590 (2018)]. The existing literature makes me think that the only novel aspect of the present manuscript is a direct verification of the connection between the two effects.

The novelty of the work is a new mechanism of phase locking in BT, which allows the gate control of the current quantization, and the concept of quantum device with new functionality. The theoretical model is developed for the operation of the BT.

The connection is by no means unexpected: in theory, the Aharonov-Casher interference sets the total amplitude of the phase slip in the device; it is the latter amplitude that determines the width and the slope of dual Shapiro steps. The authors present data that is qualitatively consistent with this expectation. The data will, certainly, be of interest to specialists that closely follow the development of “dual” superconducting electronics. In my opinion though, the advance is too incremental to merit a publication in Nature Communications, and would better belong in a more specialized journal.

We demonstrate a new mechanism of phase locking and offer quantum devices with new functionality.

In addition to the high level remark above, I have several specific questions and comments.

Questions and concerns:

– A key parameter in the theory of charge discreteness effects in a Josephson junction is the environment impedance Z (and, specifically, how Z compares to the resistance quantum $R_Q = h / 4e^2 = 6.5k\Omega$). Can the authors estimate the value of Z at relevant frequencies for their circuit? Is it simply determined by the two Pd resistors in series? Do the TiN inductances produce a significant contribution to Z ?

The impedance due to inductance at 6 GHz is $\sim 18 k\Omega > R_Q$. It is also larger than $R_{TiN}=6.3 k\Omega$.

– I don't understand the status of Eq. (4). Is this equation correct? If so, what limit does it apply in? In its present form, Eq. (4) contradicts my naive expectations. Indeed, for the case in which the phase slip amplitudes at the two junctions were the same, I would expect the relation to read $E_S = E_{S0} |\cos(\pi Q/2e)|$, and not $E_S = E_{S0} \cos(2\pi Q/2e)$. This is in full similarity to how the Josephson energy of a symmetric SQUID is $E_J = E_{J0} |\cos(\pi\Phi / \Phi_0)|$. On the other hand, if the two junctions were strongly asymmetric, I would expect $E_S = E_{S1} + E_{S2} \cos(2\pi Q/2e)$ with small oscillation amplitude, $E_{S2} \ll E_{S1}$. This does not match Eq. (4) either.

We agree with the referee, writing Eq. (4) in the form $E_S = 2E_{S0} |\cos(\pi Q/2e)|$ (symmetric case $E_{S1} = E_{S2}$) is more transparent and makes it easier to compare our results with the dual classical SQUID problem, where the critical current is modulated by magnetic flux as $I_C(\Phi) = I_C(0) |\cos(\pi\Phi/\Phi_0)|$. We have made the corresponding changes in Eq. (5). If the system is symmetric, $E_{S1} \neq E_{S2}$, then

$$E_S = \sqrt{E_{S1}^2 + E_{S2}^2 + 2E_{S1}E_{S2} \cos \frac{\pi Q g}{e}}.$$

The equation of the general form of E_S is added as Eq. (S14) to Supplementary note 3

– The authors expect that the phase slip energy in their device is $E_{S0} / h = 15$ GHz. In theory, this should correspond to the critical voltage $V_C = \pi E_{S0} / e = 200\mu V$. At the same time, the observed Coulomb blockade voltage is around $2.5\mu V$. What is the reason for such a huge gap between the observed and expected values?

We have corrected the phase slip energy, which is 0.56 GHz in each junction. It is revealed as the blockade of the current below the critical voltage. The latter is further suppressed by the noise of the TiN resistors, so that in the experiment we observe the *apparent* critical voltage (3) V_C^* .

– Currently, the main novel element of the paper – the effect of Aharonov-Casher interference on dual Shapiro steps – is confined to a single figure showing processed data [Fig. 3]. From this figure alone, it is difficult to assess the degree of tunability of dual Shapiro steps by gate voltage. Could the authors provide examples of $I(V)$ traces at different gate voltages, please (e.g., in the supplementary materials)? I think it would be very helpful if they could show $I(V)$ both at the maximum and at the minimum of the interference pattern. These data can, in principle, be used to assess how asymmetric the junctions in the authors' device are.

We are not sure about the possibility of assessing the symmetry of the JJs from the modulation of the quantized current because of presence of the parity effect. One can assess symmetry from experiments with the CQUID (S. E. de Graaf et al., Nature Physics 14, 590–594 (2018)). We conducted a few experiments with CQUID where the JJs were fabricated using the same fabrication technology as BT. The samples demonstrated high symmetry of the JJs. We add Fig. S6 to supplementary Note 5 with $I-V$ curves at different V_g . The $I-V$ curves are obtained by the numerical integration of the differential resistance.

– Even when there are no microwaves [left panel of Fig. 2], $I(V)$ has a structure with extra peaks, in addition to the main Coulomb-blockade peak. What is the origin of these extra peaks?

The screening circuit and the JJ form the resonance circuit. The peaks at the $I-V$ curve without the MW illumination are the rectification of white noise by these resonances. Thus, reduction of the inductance shifts resonances to higher frequencies.

We added text page 3, second column: “One can notice the resonances at the $I-V$ curve. The most evident one is between 2 nA and 4 nA. We relate these resonances to the rectification of the white noise by the environmental LC circuit of the BT, which has its own resonance frequencies. We confirmed that in the BT with reduced inductance, the resonances shift to higher frequencies.”

– The authors repeatedly say that the Bloch transistor is a non-dissipative element. I don't understand this statement. The authors' device has two normal state resistors. The fact that the device is dissipative is clearly exemplified by the left panel of Fig. 2 in which the dissipated power $P = I V$ is non-vanishing.

The circuit between the L_1 s is non-dissipating. One can use this branch of circuit as a source of non-dissipative quantized current.

Minor feedback:

– Parameter V_C is not defined when it is first used [see the text after Eq. (3)].

We have introduced the apparent critical voltage V_C^* in equation (3). The V_C^* is observed in the experiment.

– It is not clear from the text what thermal current δI_T means. While the authors define it in a related publication [Nature Communications 15, 9326 (2024)], but a clarification in the present manuscript would be helpful.

We gave the value of $\delta I_T \sim 1$ nA and provided an explanation in Supplementary Note 3.

– Eq. (1) in the supplement introduces the resistance R . Please define R . Is this the total resistance connected in series with the junctions?

It is correct, R is resistance in sequence.

– In the beginning of Sec. III, as well as right before Fig. 3, the authors say that the critical voltage is around $10 \mu\text{V}$. At the same time, it is clear from Fig. 2 that $V_C \approx 2.5 \mu\text{V}$. Which value is the correct one?

We corrected the typos, $V_C^* \sim 2.5 \mu\text{V}$

Typos and stylistic remarks:

– The abbreviation ISS is only used where it is introduced. Is it needed?

We remove the ISS. The sentence now reads: “In the current work we combine two devices, the CQUID and Dual Shapiro Steps, and demonstrate current quantization which can be controlled with electrostatic gating.”

– In the text, the authors introduce the critical Coulomb blockade voltage V_C . On the other hand, in the figure they use V_C^* . Is V_C^* the same thing as V_C ?

The referee is correct, there should be the apparent critical voltage V_C^* . The text is added: “There is blockade of the current without the MW below the apparent critical voltage V_C^* .”

– In Fig. 2 caption, the authors use I_A^* . Is this a typo, and should read I_C^* ?

Corrected $I_A^* \rightarrow I_C^*$

– The authors say “It is not easy to fabricate JJ smaller than $40 \times 40 \mu\text{m}^2$ in a controllable way.” This must be a typo, and they actually mean $40 \times 40 \text{ nm}^2$.

Corrected

Reviewer #4 (Remarks to the Author):

Reviewer #5 (Remarks to the Author):

Dear Editor of *Nature Communications*,

Please find the point-by-point answers (red ink) to referee questions below.

Sincerely Yours,

V Antonov

on behalf of all co-authors.

Reviewer #1 (Remarks to the Author):

1. Unfortunately, I still have concerns regarding the general style and logical flow of the article. For example, the name "Aharonov-Casher" is misspelled several times throughout the paper, despite being a central mechanism upon which Antonov et al.'s work is built. This is just one example among many in the manuscript. My intention is not to hinder the publication of this paper; rather, I strongly encourage the authors to utilize available proofreading tools to ensure that their manuscript meets the standards of Nature Communications.

We have corrected the spelling of "Aharonov-Casher effect" in the manuscript and run spellcheck on the entire text.

Reviewer #3 (Remarks to the Author):

1. I don't understand the authors' point about "the new mechanism of phase locking." Specifically, they claim that the mechanism of phase locking is different in a two-junction system as compared to a single-junction one: in a two-junction system, the phase locking is the result of oscillating charge on the island, whereas in a single-channel system the phase locking is the result of the oscillating current. In my view, presenting these as two distinct mechanisms is an overstatement. In both cases, the phase locking can be simply attributed to the oscillations of the offset charge across the junction. The offset charge itself can be induced in different ways but the fundamental

mechanism is the same. Indeed, consider the argument of the Bessel function in the case of the current bias [Eq. (4) of Nat. Comm. 15, 9326 (2024)]; what it contains is I_{ac} / f which is nothing but the offset charge.

Phase locking with the current and with the charge is substantially different in nature. The former required inductive coupling to MW, while the latter required capacitive coupling. This immediately results in a different device circuit design for optimal MW delivery. The charge phase locking allows control of the phase slip energy of BT with the polarization charge through the equations (S1)-(S12). To some extent, the charge phase locking is utilized in the Charge Quantum Interference Device (<https://doi.org/10.1038/s41567-018-0097-9>). However, the induced charge polarization in those experiments is probed in a nanosecond time interval, while in BT we are working at zero frequency.

We agree with the referee that the functional form of phase locking in case of current I_{ac} and charge dQ_g are very similar.

2. The choice of “shifting the curves for clarity” in Fig. 3(b) is a poor one in my opinion. To start with, it’s not clear which of the curves is not shifted. What’s more important is that, with the shifts, it’s very hard to get an idea of how large the gate-voltage modulation of dV/dI is compared to the overall magnitude of dV/dI . The text mentions 14% but it’s hard to see from the plots. As a reader, I’d rather prefer to see individual traces as individual panels, even if that would make the figure more bulky.

We agree with the referee that Fig. 3(b) obscures some information about the gate-voltage modulation of dV/dI . In the new Fig. 3(b), we combine individual traces dV/dI vs V_g at different quantized currents. In the absence of MW, dV/dI at the quantized plateau is ~ 20 kOhm, and the amplitude of modulation reaches 14% (right panel of Fig. S9). Under the MW, dV/dI decreases to ~ 1 kOhm, then the modulation gets 40%. We amended the text correspondingly.

3. Please define R after Eq. (3).

R is defined at the top of the second column on page 2. We added the definition after Eq. (3):
“...where R is the normal resistor of the screening circuit defined above...”

4. In the sentence “The MW control of the BT with I_{ac} and f comes from...” the parameter I_{ac} is not defined in the current version of the paper. I presume the authors ment δQ_g .

Corrected